# Molecular conservation of marsupial and eutherian placentation and lactation

**Michael W Guernsey[1], Edward B Chuong[2], Guillaume Cornelis[1], Marilyn B Renfree[3]\*[†], Julie C Baker[1]\*[†]**

[1]Department of Genetics, Stanford University School of Medicine, Stanford, United States; [2]Department of Human Genetics, University of Utah School of Medicine, Salt Lake City, United States; [3]School of BioSciences, University of Melbourne, Melbourne, Australia

**Abstract** Eutherians are often mistakenly termed 'placental mammals', but marsupials also have a placenta to mediate early embryonic development. Lactation is necessary for both infant and fetal development in eutherians and marsupials, although marsupials have a far more complex milk repertoire that facilitates morphogenesis of developmentally immature young. In this study, we demonstrate that the anatomically simple tammar placenta expresses a dynamic molecular program that is reminiscent of eutherian placentation, including both fetal and maternal signals. Further, we provide evidence that genes facilitating fetal development and nutrient transport display convergent co-option by placental and mammary gland cell types to optimize offspring success.

**\*For correspondence:** m.renfree@ unimelb.edu.au (MBR); jbaker@ stanford.edu (JCB)

[†]These authors contributed equally to this work

**Competing interests:** The authors declare that no competing interests exist.

## Introduction

While placental morphology varies widely between different mammalian species, its functions as a center for embryonic respiration, nutrient uptake, waste removal and embryonic signaling are highly conserved (*Cross, 2006*). Given this diverse set of tasks, it is intriguing how an organ with such great morphological diversity performs the same function across widely divergent taxa. For example, one critical difference in placentation across species is invasive ability. Human placental tissues invade deeply into maternal tissues where they remodel arteries, whereas other eutherian placentas, like that of the pig, do not invade (*Leiser and Kaufmann, 1994*). In marsupials, most placentas, including that of the tammar, are non-invasive (*Freyer et al., 2002*; *Freyer and Renfree, 2009*), but some, like the South American grey short-tailed opossum, do invade, albeit only for the last few days of pregnancy (*Tyndale-Biscoe and Renfree, 1987*; *Freyer et al., 2002*, *2007*). While the diversity of invasion is extreme between therian species, it is only one of the many functional, cellular, and morphological differences in placentation. This extreme diversity raises interesting questions about how a wide array of structures and cell types can mediate something as complex as fetal development.

Reproductive diversity is even more pronounced when comparing morphology and function between eutherians and marsupials, not only because the placentas are different, but also because marsupials, generally, rely more heavily on lactation to support fetal development (*Renfree, 1983*). The tammar wallaby (*Macropus eugenii*), a key model for investigating marsupial reproduction (*Renfree et al., 2011*), has a 26.5 day pregnancy followed by a 300–350 day period of lactation (*Tyndale-Biscoe and Renfree, 1987*). By comparison, the mouse has a 20 day pregnancy followed by a 20–24 day period of lactation (*Crew and Mirskaia, 1930*), relying much less on lactation for offspring success. The tammar placenta is derived from a fusion of the yolk sac and chorion, producing a structure that contains two different tissue types: the avascular bilaminar omphalopleure (BOM) and the vascular trilaminar omphalopleure (TOM). These two tissues each contain a single

**eLife digest** Before birth, mammals in their mother's womb are provided with nutrients and oxygen via an organ called the placenta. After birth, the mother produces milk to feed her young, which supports their continuing development.

The majority of living mammals, from mice to humans, belong to a group known as "eutheria" and have placentas made up of many different types of cells and tissues. This complex placenta has caused many people to wrongly refer to this group as "the placental mammals". However, marsupials like kangaroos and koalas represent a second group of mammals that also have a placenta, albeit a much simpler one that consists of only a few layers of cells.

The simpler placenta means that marsupials give birth to young that are underdeveloped compared to eutherians. These young must develop further inside the mother's pouch, where they are fed with milk that changes over time to support the different stages of their development. As a result, marsupials produce a more complicated range of milks than eutherians. To date, scientists do not fully understand how these two groups of mammals evolved to nurture their offspring in two different ways.

Guernsey et al. have now studied which genes are active in the placenta and milk-producing "mammary" glands of the tammar wallaby, a small member of the kangaroo family. Comparing the data with similar data from mice and humans revealed that marsupial and eutherian placentas are alike in many ways. For example, both rapidly change which genes are active as the placenta matures, and both contain tissues that perform specific tasks such as providing offspring with nourishment for growth and oxygen for respiration.

The analysis also shows that mammary glands in marsupials use similar genes to those used by the eutherian placenta to support the development of offspring. This suggests the placenta and the mammary gland have converged on a similar set of genes to allow female mammals to nourish their young.

Overall, the findings further scientific knowledge about the evolution of pregnancy and milk production in marsupials, paving the way for further research into the diversity of life on Earth. The findings also confirm that marsupials have working placentas, stressing that eutherians and marsupials should both be referred to as "placental mammals".

trophoblast cell layer (TL) and a single endodermal cell layer (EL) with TOM containing an additional mesodermal layer (ML) which gives rise to the vasculature (*Freyer et al., 2002*; *Renfree, 2010*) (*Figure 1*). The cells of the trophoblast layer are characterized by large nuclei, consistent with eutherian trophoblast morphology which are typically large, polyploid or multinucleated (*Zybina and Zybina, 1996*) (*Hannibal et al., 2014*), while the cells of endodermal layer contain small nuclei (*Figure 1*) (*Freyer et al., 2002*; *Jones et al., 2014*). During early pregnancy nutrient transfer to the embryo occurs across a keratinous shell coat (*Renfree, 1980*; *Menzies et al., 2011*). At day 18 the embryo loses this shell coat, and the yolk sac becomes closely interlaced with the uterine epithelium and maternal blood supply, forming the yolk sac placenta. The BOM then expands to increase the nutrient exchange surface area, while the TOM expands to further support the increased respiratory needs to the end of gestation over the next 7 days (*Renfree, 1973b*).

As the tammar fetus is underdeveloped at birth, it relies on lactation to complete its growth and development (*Trott et al., 2003*; *Green and Merchant, 1988*; *Nicholas et al., 1997*) and to provide immune protection while the adaptive immune system matures (*Joss et al., 2009*). While lactation in tammar shares some characteristics of eutherian lactation, such as initial high immunoglobulin secretion (*Old and Deane, 2000*) and nutrient density to promote growth (*Daly et al., 2007*), tammar milk undergoes dramatic and dynamic changes over the course of lactation in both type and amount of carbohydrates (*Messer and Green, 1979*), proteins, and lipids (*Green et al., 1980*; *Green and Renfree, 1982*). The composition of tammar milk is exquisitely programmed to support the remainder of fetal development, with individual proteins and amino acids secreted for differing windows of time (*Nicholas, 1988*; *Renfree et al., 1981b*; *Vander Jagt et al., 2015*) and is so potent that early pouch young fostered by a mother lactating for an older pouch young results in accelerated growth

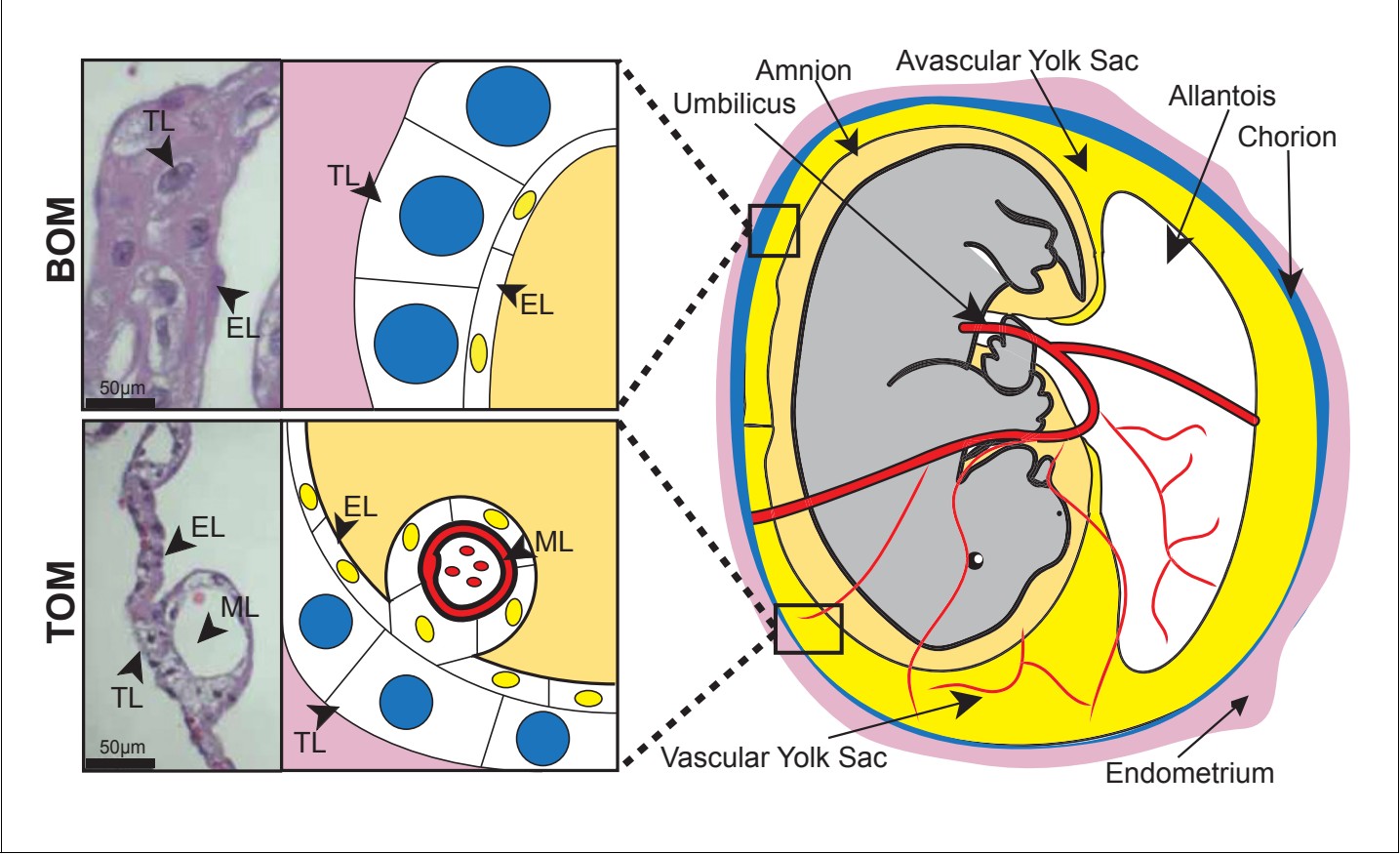

**Figure 1.** Tammar placental structure. Schematic of a day 24 tammar conceptus (right) highlighting placental cell types (middle). The tammar yolk sac placenta contains two cellular layers: the trophoblast (TL; blue) and the endoderm (EL; yellow). Further, the placental structure is divided into two halves. The 'top' non-vascularized tissue is termed the bilaminar omphalopleure (BOM) and the 'bottom' vascularized tissue is termed the trilaminar omphalopleure (TOM) which also contains a mesodermal layer (ML). H&E sections (left) demonstrate the cell types within the BOM and TOM components (Scale bars are 50 μm).

and development, including: increases in body size, head length, thicker fur, and accelerated maturation of the fore-stomach (*Nicholas et al., 1997*); Menzies et al., 2007; *Kwek et al., 2009*). To date, the signals within the milk that drive this acceleration are unknown, but the milk appears to be crafted to promote fetal morphogenesis in specific windows of time (*Green et al., 1980*), which is related to variation in both resource availability in the environment (*Dennis and Marsh, 1997*) and neural cues from the mother (*Renfree MB et al., 1981a*). As marsupial milk is clearly supplied to support fetal development, a role fulfilled primarily by the placenta in eutherians, it has been proposed that dynamic and extended lactation periods evolved in marsupials as an alternative to the complex placentation seen in eutherians (*Renfree, 1983*; *Tyndale-Biscoe and Renfree, 1987*; *Renfree, 2010*).

To better understand the evolution of pregnancy and lactation in marsupials, we generated and analyzed transcriptome data from placental and mammary gland tissues of the tammar wallaby. We find that the tammar yolk sac placenta shares striking molecular similarity to the chorioallantoic placenta of eutherians, despite major morphological and developmental differences. Moreover, we find that established protein markers that distinguish decidua (maternal) and trophoblast (fetal) tissue in the eutherian placenta are instead co-localized in the yolk sac placenta, suggestive of an ancestral molecular program underlying placentation that has undergone further compartmentalization during eutherian evolution. Furthermore, we provide additional molecular confirmation that marsupials do have fully functional placentas, which supports the growing body of evidence that eutherians and marsupials can both be classified as 'placental mammals' (*Padykula and Taylor, 1982*; *Freyer et al.,*

*2003*; *Renfree, 2010*). Finally, we discover that genes crucial for eutherian placental function are expressed in the tammar mammary gland. This provides the first comprehensive molecular evidence that marsupials favored a physiologically complex lactation to achieve the same developmental milestones that placentation has facilitated in eutherians.

## Results

### Dynamic changes occur during gestation in the tammar placenta

To investigate how the marsupial placenta functions we used 3SEQ (*Beck et al., 2010*) to sequence the BOM and TOM transcriptomes at day 21–23 and day 25. To examine the specific molecular characteristics of the BOM and TOM components of the placenta, we pooled BOM (4 replicates) and TOM (6 replicates) sequences from all gestational time points. Using DESeq2, we find 445 differentially expressed transcripts; of which 83 were up-regulated in BOM tissue and 362 were up-regulated in TOM tissue (*Figure 2A*). The BOM up-regulated genes are associated with ontology related to endocytosis and metabolism (p-adj = 0.0764, 0.0390, see *Supplementary file 1*) consistent with the hypothesis that BOM acts as the center for embryonic nutrient uptake (*Renfree, 1973b*). Conversely, the genes up-regulated in TOM are associated with ontology related to extracellular matrix organization, hematopoiesis, and response to oxygen levels (p-adj = $1.49 \times 10^{-7}$, 0.00193, 0.0688, see *Supplementary file 1*). This is consistent with the idea that TOM, being the vascularized component

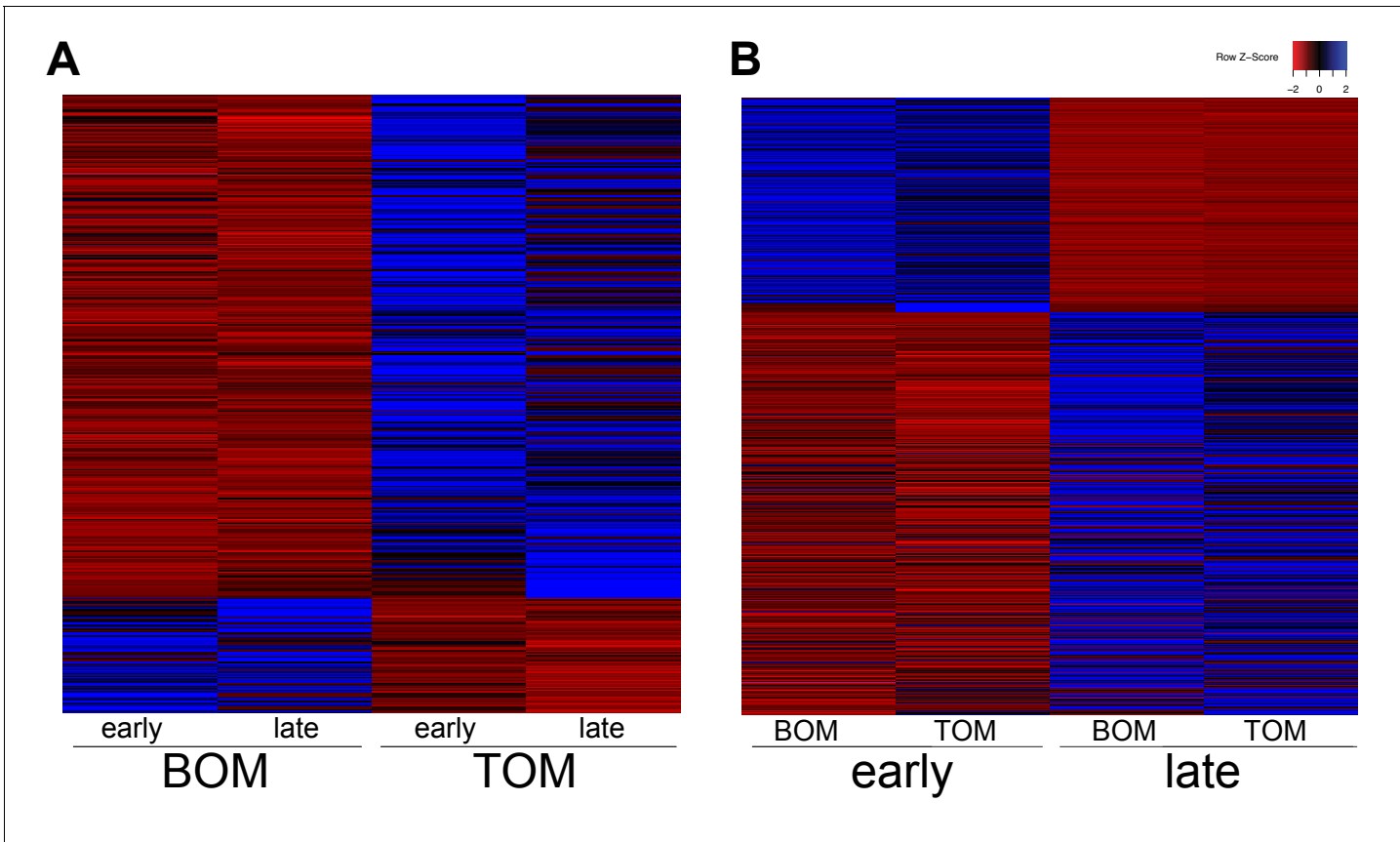

**Figure 2.** Tammar placental gene expression differs between tissues and over time. (**A**) Transcripts differentially expressed between BOM and TOM (n = 445) with blue indicating high expression and red low expression (see Z-score insert upper right). (**B**) Transcripts differentially expressed between early (day 21–23) and late (day 25) placental time points (n = 1705).

The following figure supplement is available for figure 2:

**Figure supplement 1.** Tammar placenta transcriptome exhibits distinct signatures in both tissue type and time.

of the tammar placenta, acts as the center of embryonic respiration (*Renfree, 1973b*). Overall, this supports the suggestion that BOM and TOM have specific and distinct functions throughout gestation (*Renfree, 1973b*) (*Figure 2—figure supplement 1A*), consistent with previously observed morphology differences (*Freyer et al., 2002*).

After the shell coat is lost by gestational day 18, TOM begins to expand through day 23 of gestation until it has covered 50% of the yolk sac (*Freyer et al., 2002*). We examined if the function of the placenta changed from day 21–23, when TOM is still expanding, to day 25, when TOM has finished expanding. To this end, we pooled the 21–23 day sequences (4 replicates) together and the 25 day sequences (6 replicates) together, including both BOM and TOM. Using DESeq2, we find 1705 differentially expressed transcripts; of which 592 were up-regulated at day 21–23 (early) and 1113 were up-regulated at day 25 (late) (*Figure 2B*). Genes up-regulated at earlier time points exhibited no significant enrichment for any gene ontology, but trended toward intracellular and ion transport (p-adj = 0.188, 0.109, see *Supplementary file 1*). In contrast, those up-regulated at later time points were enriched for cellular growth, differentiation, and junction organization (p-adj = 0.0331, 0.0162, $5.79 \times 10^{-4}$, see *Supplementary file 1*). Overall, these findings reveal that the tammar placenta undergoes dynamic changes in gene expression throughout its development despite the short duration of its physiological activity (*Figure 2—figure supplement 1B*).

## Tammar placenta expresses eutherian molecular signatures

Although the overall morphology and cellular structure of the chorioallantoic placenta in mouse and humans is vastly different from the yolk sac placenta in marsupials, we next tested whether they shared similarity at the molecular level. To this end, we compared the tammar placenta transcriptome, pooling samples of all tissue types and time points, to publically available traditional RNA-Seq datasets for the term mouse (ENCSR000BZP) and human (GSE56524) placenta transcriptomes. We found 3894 transcripts shared by all three species (*Supplementary file 2*) which included many factors that have documented roles in placental function (*Rawn and Cross, 2008*). We uncovered the expression of trophoblast markers including: *Gcm1* (*Anson-Cartwright et al., 2000*), members of the *Igf* signaling pathway (*Reik et al., 2003*), many placental *cytokeratins* (*Gauster et al., 2013*), *Wnt7b* (*Parr et al., 2001*), *Stra13* (*Hughes et al., 2004*), *PTN* (*Schulte et al., 1996*), and *Gjb3* (*Koch et al., 2012*). *Cathepsins* (*Mason, 2008*) and *Serpins* (*Kaiserman et al., 2002*) a class of proteases and protease inhibitors respectively, which are important for mediating maternal fetal interactions were also present (*Hannibal and Baker, 2016*). We also found markers of the maternal decidua such as *Cebpb* (*Lynch et al., 2011*) and *Vim* (*Can et al., 1995*). Additionally, the tammar and mouse placentas share the expression of *Cdx2*, a transcription factor that is essential for early trophoblast development (*Strumpf et al., 2005*; *Frankenberg et al., 2013*), while the tammar and human placentas share expression of *Nodal*, a secreted factor important for trophoblast differentiation (*Ma et al., 2001*). These findings are consistent with the idea that a core set of genes is essential for placental function and development across all therian mammals and that, regardless of morphological differences, both marsupial and eutherian placentas share expression of key placental transcripts.

Eutherian and marsupial placentas have key differences in their cellular composition: the eutherian placenta is derived from both fetal trophoblast and maternal decidua cells, and the marsupial placenta is derived solely from fetal trophoblast and yolk sac endoderm cells. Given our observation that the tammar placenta expresses genes that function in eutherian placentation, we next examined where these proteins are expressed at the cellular level. To this end, we performed immunofluorescence using antibodies against the trophoblast markers - CDX2, GCM1, and Pan-Cytokeratins and the maternal decidual markers - CEBPB and Vimentin - on day 24 tammar BOM (*Figure 3*) and TOM (*Figure 3—figure supplement 1*) tissues. We find that CDX2, CEBPB and Vimentin are expressed in both the endodermal and trophoblast layers, with CDX2 and CEPBP being nuclear and Vimentin cytoplasmic. This is particularly intriguing as CEBPB (*Lynch et al., 2011*) and Vimentin (*Can et al., 1995*) are known markers of maternal decidua in eutherians and have limited expression in the eutherian trophoblast. Additionally, we find that the known eutherian markers for trophoblasts, Pan-Cytokeratins (*Gauster et al., 2013*) and GCM1(*Anson-Cartwright et al., 2000*) are strongly expressed only in the endodermal layer, and are not expressed in the trophoblast layer. Overall, protein localization in tammar reveals substantial findings including the expression of maternal decidual markers throughout the yolk sac placenta and fetal trophoblast markers expressed exclusively within

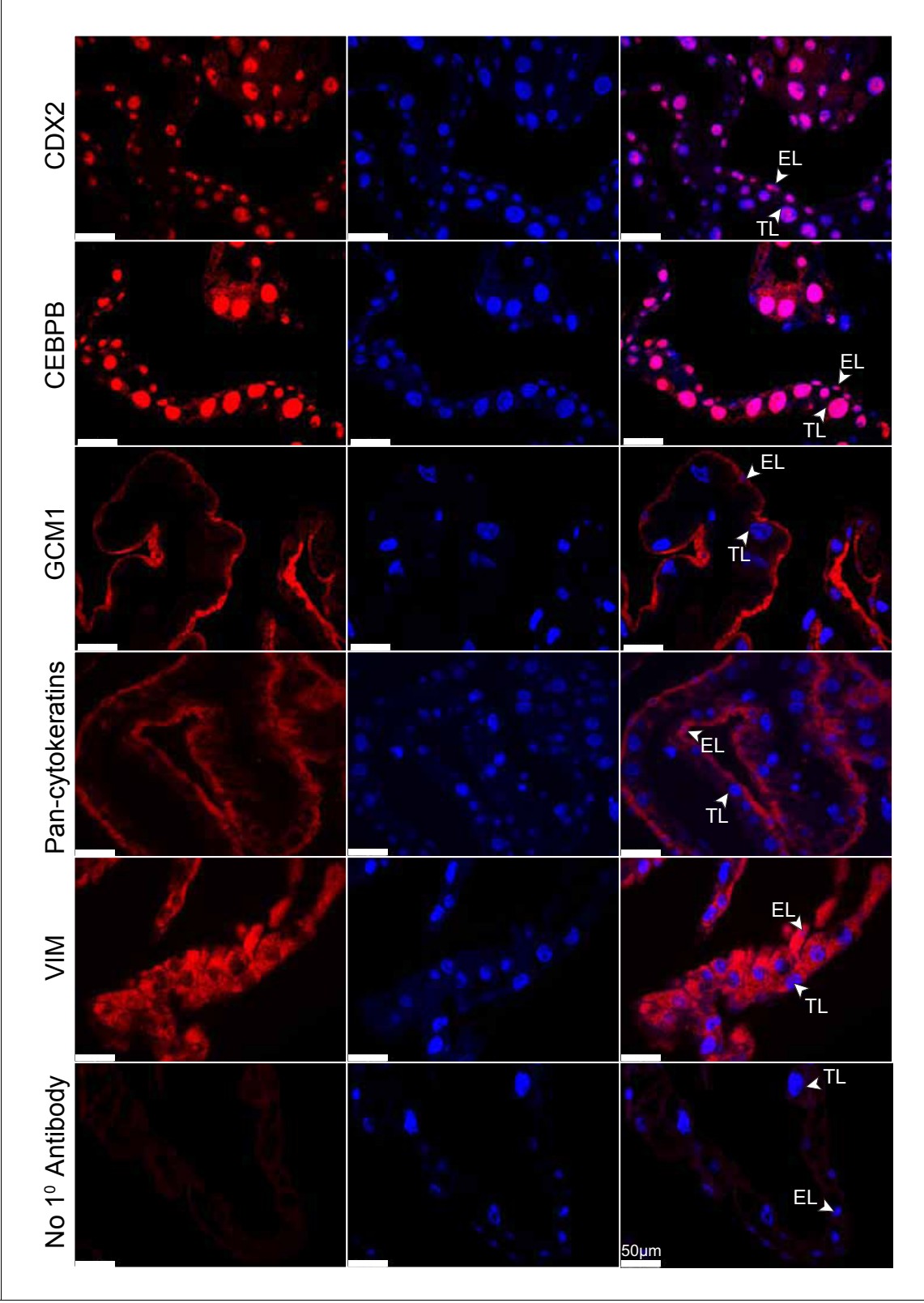

**Figure 3.** Tammar placenta expresses eutherian placental markers. Immunofluorescence of the eutherian placental markers: CDX2, CEBPB, GCM1, Pan-cytokeratins, and Vimentin, in day 24 tammar placenta. The left column shows protein expression. The middle is DAPI stain depicting the nucleus for the same section. The right column is the merge. No 1° Antibody is a negative control (bottom row). Scale bars are 50 μm.

*Figure 3 continued on next page*

*Figure 3 continued*

The following figure supplement is available for figure 3:

**Figure supplement 1.** Tammar placenta expresses eutherian placental markers.

the endoderm. Taken together, these data suggest that the choriovitelline placenta in tammar has accomplished coordination of critical placental functions and may even include functions that are supplied maternally in eutherians.

## Tammar placenta expresses early eutherian molecular program

We next examined whether the similarity between tammar and eutherian placentation exist during both the early and late stages of placentation. While we demonstrate that the tammar placental transcriptome shares molecular similarities to the term eutherian placenta, the mouse and human placenta are molecularly distinct between the first half and second half of pregnancy, with the early stages primarily involved in cell cycle and metabolism and the later stages focused on reproduction (*Knox and Baker, 2008*; *Winn et al., 2007*). Therefore we tested whether the tammar placenta might share greater similarity with a specific stage of eutherian placentation. To this end, we compared the transcriptome of all sequenced tammar time points with gene expression in the mouse placenta throughout gestation (e8.0-term) (GSE11220), and to the transcriptome of the adult mouse heart (ENCFF204IFN). We identified the transcripts expressed in both the tammar and mouse, ranked the expression of each transcript at each time point from highest to lowest, and compared the time points using a series of pairwise Spearman correlation tests. The highest coefficients reveal that the transcriptome of the tammar placenta, regardless of gestational age, is more similar to that of an e10.5 mouse placenta than any other time point (*Figure 4A*). Additionally, we find that the lowest coefficients exist between the day 21 and 23 tammar placenta and the later stages of mouse development (e17.0, e19.0, and birth) (*Figure 4A*). Finally, we find that the mouse placenta has higher correlations at all time points with the tammar placenta than it does with the mouse heart (*Figure 4—figure supplement 1*). This suggests we are detecting a true conservation of placenta-specific transcription and not just species-specific transcription. While the Spearman coefficients are higher between the mouse e10.5 and tammar transcriptomes than in any other pairwise comparison, this difference is minimal and therefore we next sought to refine the analysis by comparing only placenta-specific transcripts, removing housekeeping and other transcripts that may contribute to noise. In a previous study, we identified 410 transcripts are unique to the mouse placenta (*Knox and Baker, 2008*). During gestation 340 of these transcripts are expressed prior to e12.5 and 70 are expressed after e12.5 of mouse gestation, demarcating an early and late placental signature (*Knox and Baker, 2008*). We found 100 tammar orthologs of these 410 mouse placental transcripts expressed in our datasets at 21, 23 and 25 days gestation. Of these 100, 90 are expressed only in the early mouse placenta (prior to 12.5) and 10 are expressed only in the late mouse placenta (after 12.5) (*Figure 4B*). When comparing the early transcripts (90/340) to the late placental transcripts (10/70), we find a significant enrichment of early transcripts ($p<0.05$ by chi-square test). Overall, this more targeted analysis supports the idea that tammar placentation functionally resembles the earlier phases of mouse placentation.

## Tammar mammary gland and eutherian placenta share molecular functions

The evolution of complex lactation in marsupials represents an independent strategy that has been proposed to mirror aspects of eutherian placentation (*Renfree, 1983*, *2010*). Previously, it was demonstrated that *Igf2* was imprinted in the tammar mammary gland (*Stringer et al., 2012*, *2014*), a phenomenon previously determined to be essential for proper growth and development of the eutherian placenta (*Reik et al., 2003*). This suggests that the tammar mammary gland may indeed express similar transcripts as the placenta. To identify any conservation of function between organ systems, we first performed transcriptome sequencing using 3SEQ on tammar mammary glands during early lactation periods (days 36, 60, and 95). We next compared the mammary gland transcriptomes with the transcriptomes of tammar placenta, tammar liver, tammar testis, mouse placenta,

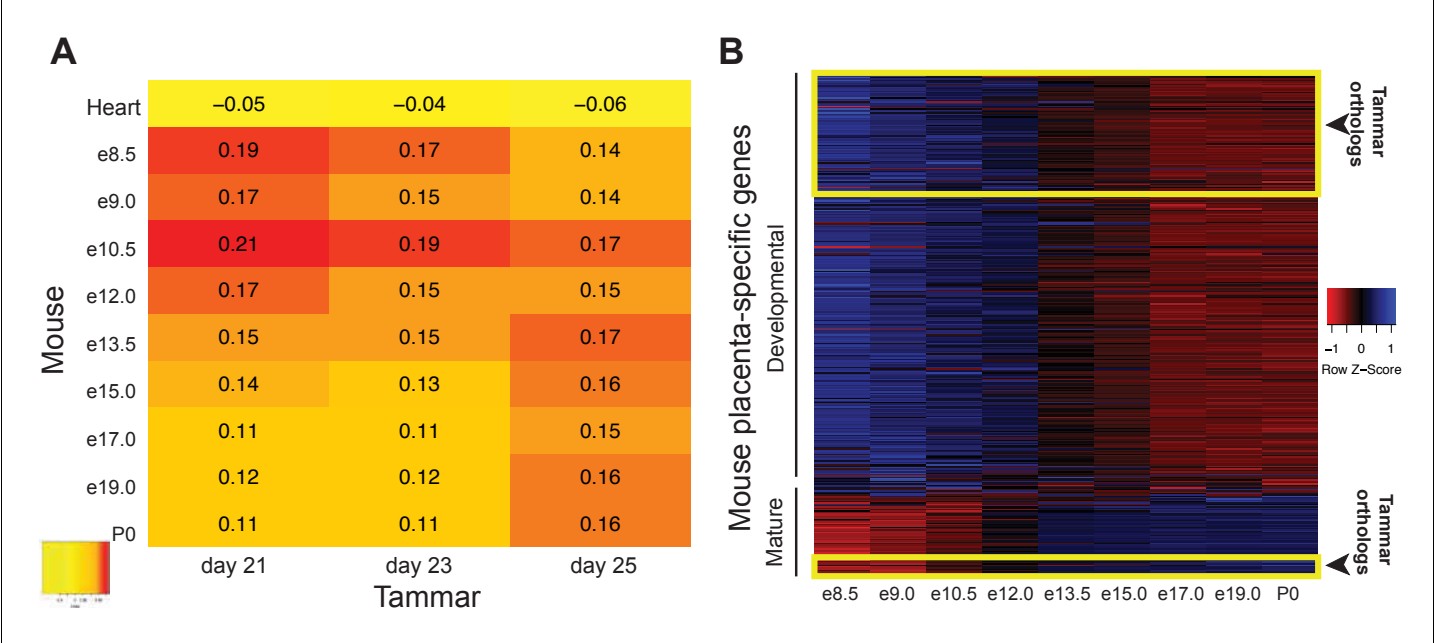

**Figure 4.** The tammar placenta is more similar to the midgestation mouse placenta. (**A**) Comparison of mouse and tammar placental transcriptomes at different gestational ages using Spearman correlations (see values throughout graph). Red indicates the greatest level of transcriptional similarity while yellow indicates the weakest transcriptional similarity. The mouse heart transcriptome used as a control (top). (**B**) Heat map depicting the expression of 410 transcripts that are specific for the mouse placenta of which 340 are expressed before and 70 are expressed after midgestation (e12.5). The tammar placenta expresses orthologs of 100 of these transcripts, which are highlighted in the yellow boxes on the heat map. 90 are classified as developmental, or expressed early in mouse, and 10 are classified as mature, or expressed late in mouse. Blue indicates high expression and red low expression (see Z-score on right).

The following figure supplement is available for figure 4:

**Figure supplement 1.** Tammar placenta exhbits conservation of placenta-specific gene expression.

mouse liver, mouse testis, and mouse mammary gland. After ranking the expression of shared transcripts and calculating pairwise Spearman correlations, several findings stand out. First, we find that the tammar and mouse placenta share the highest correlation of any cross-species organ pair (**Figure 5—figure supplement 1**), confirming the high degree of molecular similarity between the eutherian and tammar placenta that we found using immunofluorescence (**Figure 3**). Second, although less so than the placenta, the tammar mammary gland shows the highest correlation with the mouse mammary gland when compared to any other mouse tissue, highlighting some functional conservation in this organ. Among those genes with conserved expression in mammary glands, we found *Gata3* (**Chiu and Chen, 2016**) and *Msh2* (**Tanaka et al., 1997**), transcription factors known to be essential for placentation in mice. Immunofluorescence in tammar mammary gland at day 60 of lactation finds GATA3 in the alveoli (**Figure 5B**). This pattern of expression is reminiscent of GATA3 expression in the mouse mammary gland (**Tlsty, 2007**), suggesting an ancestral role of GATA3 in mediating lactation. Finally, the placentas of both tammar and mouse show the lowest within species correlation with mammary gland which suggests widely divergent roles for each organ in supporting reproductive physiology.

Although the transcriptional repertoire is globally distinct between placenta and mammary gland in both tammar and mouse we next examined whether there were genes expressed in both organs that might reflect any shared functions. First, we tested whether placental transcripts could be found specifically in the tammar mammary gland, suggesting a role in supporting tammar fetal morphogenesis. To this end, we identified transcripts present in eutherian placenta (**Supplementary file 1**), tammar placenta, tammar mammary gland, but not in the mouse mammary gland (GSE60450),

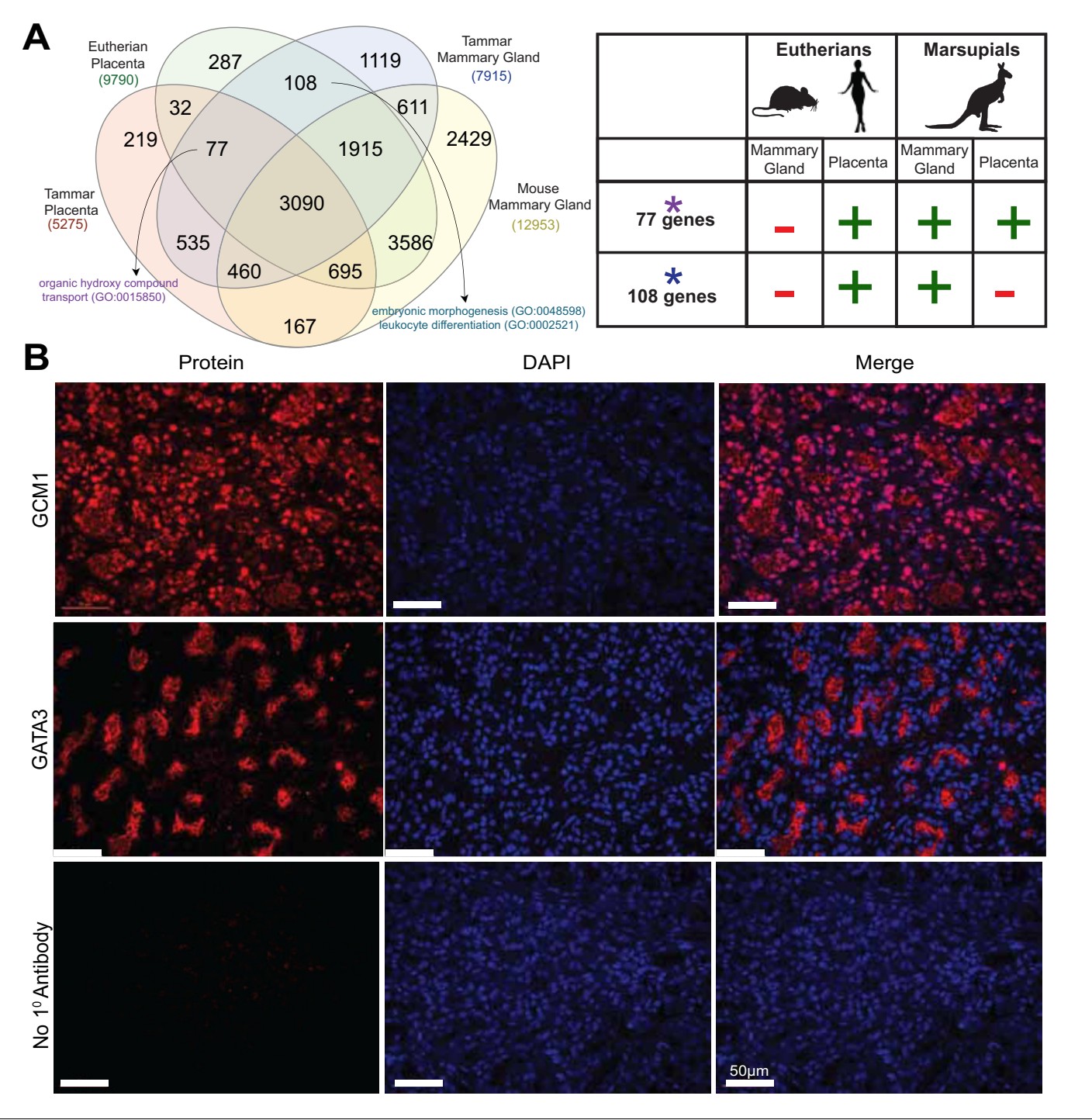

**Figure 5.** Shuffling of reproduction genes between lactation and placentation. (**A**) Venn-diagram (Left) comparing the genes expressed in the lactating tammar and mouse mammary glands with the tammar and eutherian placenta. Key ontology of genes in overlapping categories is highlighted. Genes listed for each category in **Supplementary file 3**. The table (Right) depicts key categories of genes from the venn-diagram and whether or not they are expressed in the placenta and mammary gland tissues of the tammar and eutherians. If the gene class is present in the tissue of a given lineage it is given a green '+', if it is absent it is given a red "-". (**B**) Immunofluorescence of GCM1 and GATA3. The first column depicts the expression of each protein, the second column is DAPI stain for nuclear DNA, and the third column is a merge of both images. The final row contains a negative control where no primary antibody was applied (Scale bars are 50 μm).

The following figure supplement is available for figure 5:

*Figure 5 continued on next page*

*Figure 5 continued*

**Figure supplement 1.** Tammar placenta and mammary gland have exceptionally distinct molecular functions.

finding 77 genes expressed. As we could find genes simply by chance that meet these criteria, we also compared placental transcripts to other organ systems in a similar fashion, including tammar liver, mouse liver, tammar testis, and mouse testis. While we find gene expression shared in all categories, including 76 genes shared between tammar placenta, eutherian placenta, tammar liver, but not mouse liver, and 589 genes shared between tammar placenta, eutherian placenta, tammar testis, but not mouse testis, we only find significant functional associations when comparing placenta to mammary gland. The 77 genes shared between the tammar placenta, eutherian placenta, and tammar mammary gland (*Figure 5A*), are associated with nutrient transport (GO analysis; p-adj = 0.0163, see *Supplementary file 3*) and include the placental transcription factor *Gcm1*, which is necessary for eutherian placentation (*Anson-Cartwright et al., 2000*). Localization of GCM1 by immunofluorescence demonstrates strong nuclear staining throughout the tammar mammary gland at day 60 of lactation. Additionally, we find high expression of GCM1 in the alveoli (*Figure 5B*), the site of milk ejection in the tammar mammary gland (*Findlay, 1982*). This expression suggests a novel function of GCM1 in supporting multi-stage lactation in marsupials. Therefore, we suggest that these 77 transcripts, including *Gcm1*, represent genes that have been convergently co-opted by the marsupial mammary gland to expand its potential for fetal support or lost in the eutherian mammary gland where the placenta provides all the required support for fetal development.

We next tested whether there were transcripts shared only between eutherian placentas and the tammar mammary gland suggesting conservation of genes used for late fetal development in two different reproductive organs. To this end, we identified 108 transcripts that were expressed in the eutherian placenta and the tammar mammary gland, but not in the tammar placenta and mouse mammary gland (*Figure 5A*). As we could find genes simply by chance that meet these criteria, we again compared placental transcripts to the other organ systems used in our previous analysis. While we find gene expression shared in all categories, including 168 genes shared between eutherian placenta and tammar liver, but not tammar placenta and mouse liver, and 589 genes shared between eutherian placenta and tammar testis, but not tammar placenta and mouse testis. GO analysis of these 108 genes did not unveil significant ontology, but trended toward processes critical for reproductive success including: embryonic morphogenesis and immune function (p-adj = 0.163, 0.146, see *Supplementary file 3*). Among the genes associated with these two GO terms we find *Dkk1* and *Csf1r*, transcripts with documented roles in placentation (*Pollheimer et al., 2006*; *Sasmono et al., 2003*). Here we also find the expression of *Igfbp1*, a molecule important for regulating insulin growth factor function, a process previously demonstrated to be important in both therian placentation (*Reik et al., 2003*) and marsupial lactation (*Stringer et al., 2012*). Conversely, the liver and testis analysis does not reveal any significant GO enrichment associated with placentation. Together, these data provide molecular evidence that the tammar mammary gland performs some similar functions to the late eutherian placenta to support continued fetal development in the pouch.

## Discussion

### The tammar placenta shares conserved function with the eutherian placenta

The yolk sac structure of the tammar placenta has led to the idea that it does not function like the eutherian placenta and marsupials are therefore often misleadingly termed 'non-placental'. However, we find that the tammar yolk sac placenta expresses many key markers of eutherian placental cell types, including those of maternal decidua. This suggests that the tammar yolk sac placenta perform functions that the eutherian placenta has compartmentalized into more complex cellular domains. Additionally we find that global gene expression undergoes dynamic shifts from day 21 and 23 to day 25, during tammar gestation, a window critical for placental growth and proliferation. This same type of dynamic change in placental gene expression has also been documented in the mouse (*Knox and Baker, 2008*) and human (*Winn et al., 2007*) suggesting rapidly changing gene

expression patterns may be conserved between therian placentas. The presence and expression of core placental genes and the dynamic gene expression in the tammar placenta suggests that many molecular functions of the placenta may be conserved in the therian mammal stem species, as suggested previously (*Freyer et al., 2002*, *2007*). Furthermore, this strongly suggests that the use of the placenta to distinguish eutherians from marsupials is incorrect.

## Functional convergence of the placenta and mammary gland in the evolution of therian mammals

We suggest that reproductive genes have been shared between the placenta and the mammary gland throughout evolution to establish diverse lactation and placentation strategies to nourish mammalian young. We show that these shifting events have occurred both to facilitate the multistage lactation of marsupials and to allow for the formation of placentas of diverse cell and tissue types that allow for longer pregnancy in eutherians. The genes shared between these tissues are enriched for processes involved in nutrition, immune function and embryonic morphogenesis suggesting a functional convergence between placentation and lactation. Our finding that the mammary gland of the tammar shares the expression of transcripts, including *Gcm1* (*Anson-Cartwright et al., 2000*) and *Gata3* (*Chiu and Chen, 2016*), that promote placental functions in eutherians suggests marsupial lactation is an alternate strategy that parallels eutherian placentation. Taken together these data support the idea that reproductive genes that allow efficient exchange between mother and offspring have been respectively co-opted in placentation and lactation to facilitate proper offspring growth and development.

## Molecules provide novel insights into marsupial placentation

While anatomical work has described the tammar placenta, our data provides a molecular complement. We first confirm (*Renfree, 1973b*) that BOM and TOM are discrete placental structures with independent functions. Many of the genes up-regulated in BOM fall into ontologies that are associated with nutrient uptake and metabolism. This is consistent with the hypothesis that BOM, with its large trophoblast layer, has a high cytoplasm-to-nucleus ratio allowing for greater uptake and subsequent metabolism of nutrients from uterine secretions (*Renfree, 1973b*; *Renfree and Renfree, 1973a*; *Renfree and Tyndale-Biscoe, 1973*; *Freyer et al., 2002*). Many of the genes up-regulated in TOM fall into ontologies that are associated with respiration. This is consistent with the idea that TOM, having vasculature and a thin trophoblast layer, allows for rapid transport of oxygen thus acting as the primary site of embryonic respiration (*Renfree, 1973b*; *Renfree and Renfree, 1973a*; *Renfree and Tyndale-Biscoe, 1973*; *Freyer et al., 2002*). This is especially intriguing because the thinning of the trophoblast layer in TOM is reminiscent of the convergent thinning of the maternal-fetal interhemal distance in eutherian placentation (*Enders and Carter, 2004*), suggesting an additional layer of morphological convergence.

In addition, our work provides strong molecular support for the earlier suggestions that the endoderm and trophoblast layers of the tammar placenta have each adopted separate and critical placental functions. Based on expression of both GCM1 and Pan-cytokeratins, we suggest that the endodermal layer plays a key role in tammar placenta function. These genes mark critical components of the trophoblast (or fetal) component of a wide variety of eutherian placentas (*Enders et al., 2006*). GCM1 specifically marks the placental labyrinth (*Anson-Cartwright et al., 2000*), the site of nutrient uptake from maternal blood (*Cross, 2006*), suggesting that the endodermal layer is serving as a center of nutrient trafficking in the tammar placenta, a function thought to be held by the trophoblast layer. Because there is no decidualization in the tammar, it also is surprising to find CEBPB and Vimentin in the tammar trophoblast layer, as these are known decidua (or maternal) markers in eutherians. Additionally, CEBPB is a direct regulator of decidualization in eutherians (*Wagner et al., 2014*). Further comparative studies will need to be performed to determine whether the tammar trophoblast layer gained maternal placenta function or whether eutherians expanded fetal placenta function to include the decidua. Overall, the layer-specific expression of these proteins suggests that the endodermal and trophoblast layers of the tammar placenta play similar roles to the eutherian trophoblast and decidua tissues, respectively.

## Conclusion

Our findings indicate that, despite its anatomical simplicity, the tammar placenta expresses a dynamic molecular program that is highly reminiscent of the eutherian placenta. Additionally we provide evidence that genes underlying important organismal functions may move freely between different cell and tissue types throughout the course of morphological evolution. This is highlighted in both the sub-functionalization of the tammar placenta to express the genes of both the maternal and fetal placenta tissues of eutherians in distinct cell layers and by the convergent co-option of key reproductive transcripts for use in placentation and lactation throughout mammalian evolution. Overall, our study highlights the molecular conservation in mammalian placentation despite the enormous morphological variation and provides some of the first molecular data on how the marsupial lineage fits into this framework.

# Materials and methods

## Samples

Pouch young were removed (RPY) from adult female tammars to reactivate their diapausing blastocysts. Pregnant females were euthanized during the last third of gestation to collect placental tissues from days 21, 23, 24, and 25 RPY. Mammary gland tissue was collected from females carrying pouch young at days 36, 60, and 95 post partum, during phase 2A of lactation (*Green and Merchant, 1988*).

## 3' RNA-seq (3SEQ)

3SEQ captures only the sequence of the 3' end of transcripts, yielding a single read per transcript, allowing for quantification of expression while using fewer total reads (*Finn et al., 2014*). For placenta tissue, one sample of day 21 bilaminar omphalopleure (avascular yolk sac) (BOM), day 21 trilaminar omphalopleure (vascular yolk sac)(TOM), day 23 BOM, day 23 TOM, two samples of day 25 BOM, and four samples of day 25 TOM liquid nitrogen frozen tissues were prepared for sequencing. mRNA was isolated using Dynabeads Oligo (dT)$_{25}$. 3SEQ libraries were prepared from this mRNA as previously described (*Chuong et al., 2013*). In brief, the samples were given a short heat-shearing treatment immediately followed by cDNA synthesis. The resulting cDNA was repaired using 3'A-tailing and ligated to linkers containing unique Illumina barcodes for sequencing. An E-gel SizeSelect agarose gel was used for size selection and the samples were then PCR amplified for 15 cycles and purified using AMPure XP beads. Quality of the libraries was assed using both Qubit and Bioanlyzer technology, which were then sequenced on the Genome Analyzer IIx.

For mammary glands, a single sample from each of day 36, 60, and 95 paraformaldehyde fixed tissues were prepared for sequencing. 3SEQ proved especially useful in this experiment because it is known to be more effective than traditional RNA-seq when used on fixed or archived tissue samples (*Beck et al., 2010*). Fixed samples were initially subjected to a brief protease digestion to remove protein to nucleic acid cross-linking. Total RNA was then extracted using the ambion RecoverAll Total Nucleic Acid Isolation kit and mRNA was subsequently isolated using Dynabeads Oligo (dT)$_{25}$. 3SEQ libraries were prepared from this mRNA as previously described (*Chuong et al., 2013*). In brief, the samples were given a short heat-shearing treatment immediately followed by cDNA synthesis. The resulting cDNA was repaired using 3'A-tailing and ligated to linkers containing unique Illumina barcodes for sequencing. A 3% NuSieve GTG agarose gel was used for size selection and the samples were then PCR amplified for 17 cycles and purified using AMPure XP beads. Quality of the libraries was assed using both Qubit and Bioanlyzer technology, which were then sequenced on the Illumina NextSeq.

The resulting sequences were aligned to the tammar wallaby genome (m.eug_v1.0) (*Renfree et al., 2011*) using STAR (*Dobin et al., 2013*) (version 2.5.1b, RRID:SCR_005622). Raw counts were assigned to significantly transcribed regions using UniPeak (*Foley and Sidow, 2013*) v1.0. Regions were then associated with the nearest gene using HOMER (*Heinz et al., 2010*) (version 4.7, RRID:SCR_010881). Multiple regions mapping to the same gene were combined to give a final read count for each gene. The resulting data was then normalized across samples and tissue types using the bioconductor package DESeq2 (*Love et al., 2014*) in R (RRID:SCR_000154). The data files

processed in this study were deposited at the Gene Expression Omnibus (GEO) accession number GSE90838.

## Differential expression analysis

Differentially expressed transcripts were identified using two pairwise comparisons. First we compared placental tissue types (BOM vs. TOM) and then we compared gestational time points: time during TOM expansion (day 21 and 23, termed 'early') vs. time after TOM expansion (day 25, termed 'late'). While some samples had a single replicate sequenced our strategy of grouping the resulting data by both time and tissue type allowed us to achieve significant statistical power. Differential expression was determined using the DEseq2 bioconductor package in R with any transcripts significantly up or downregulated (Benjamini-adjusted $p<0.05$) being included in subsequent analyses. The Ensembl transcript names were used to obtain associated gene names from Ensembl biomart. The resulting gene lists for each comparison were entered into Enrichr (*Kuleshov et al., 2016*) for gene ontology (GO) analysis to assess any relevant differences in biological function picked up by our comparisons.

## Comparative genomics analysis

To compare our newly assembled tammar placenta and mammary gland transcriptomes to those of eutherian mammals we used several publically available gene expression datasets. RNA-seq based transcriptome data for the term mouse placenta was obtained from ENCODE (ENCSR000BZP). RNA-seq based transcriptome data for the human term placenta was obtained from the gene expression omnibus (GEO) (GSE56524) (*Metsalu et al., 2014*). RNA-seq based transcriptome data for mouse lactating luminal and basal cells of the mammary gland was obtained from GEO (GSE60450) (*Zhou et al., 2014*). Read count tables were obtained for all of these data sets and the average number of reads per transcript was taken, after this those transcripts that were below 1% of the mean were excluded and dubbed not expressed. For the mouse lactating mammary gland transcriptomes the basal and luminal cells were combined to get the best representation of total mammary gland function. The remaining genes were dubbed expressed and used to compare against the tammar transcriptomes. Resulting gene lists for meaningful comparisons were entered into Enrichr (*Kuleshov et al., 2016*) for gene ontology (GO) analysis to assess any relevant differences in biological function picked up by our comparative transcriptomics.

To compare gene expression throughout placental development we compared our tammar placenta transcriptome time course to microarray based transcriptome data documenting gene expression in the developing mouse placenta from GEO (GSE11220) (*Knox and Baker, 2008*). Additionally, we compared our data to ENCODE adult mouse heart transcriptome data (ENCFF204IFN) to be sure we were detecting conservation of organ-specific expression and not just species-specific expression. To make these data sets comparable we reduced the data to include only those genes expressed in the developing tammar and mouse placentas and the adult mouse heart. From here we ranked the expression of these genes in each time point of each species from highest to lowest and performed a series of pairwise Spearman correlations to assess the degree of similarity between each. The resulting Spearman correlation coefficients comparing tammar and mouse were then used to assess where the greatest transcriptional similarities exist. To compare the transcriptional similarity of the mammary glands and placenta we used the same approach. We compared our tammar placenta and mammary gland transcriptomes to RNA-seq based transcriptomes from GEO for the tammar liver, testis (GSE50747), and mouse mammary gland (GSE60450) and Encode data for the mouse placenta (ENCSR000BZP), liver (ENCSR216KLZ, ENCSR000BYS), and testis (ENCSR266ESZ, ENCSR000BYW). Liver and testis tissues were included to ensure that any relationship detected between the placenta and mammary gland are unique and not a relationship that occurs when comparing any two organs.

## Immunofluorescence

Samples for immunofluorescence (IF) were fixed overnight at 4°C in a 4% paraformaldehyde solution. They were then processed while rocking at room temperature through five 10 min washes in PBS, one 1 hr wash in 70% ethanol, one 1 hr wash in 85% ethanol, one 1 hr wash in 90% ethanol, one 1 hr wash in 95% ethanol, three 20 min washes in 100% ethanol, and three 10 min washes in xylenes.

Samples were then placed at 65°C in paraffin for 1 hr, the paraffin was then replaced and left overnight, and the tissues were embedded in paraffin and put into blocks the following day. Sections were then taken from these blocks on a Leica RM 2155 at a thickness of 5 µm and mounted on glass slides and left to dry overnight at 37°C.

Slides for immunofluorescence were processed as follows: two 10 min washes in xylenes, two 10 min washes in 100% ethanol, two 10 min washes in 95% ethanol, two 10 min washes in 80% ethanol, two 10 min washes in 70% ethanol, one 10 min wash in 50% ethanol, and two 10 min washes in deionized water. The protocol for GCM1 on mammary gland tissue included antigen retrieval in sodium citrate buffer (pH 6.0) for 13 min in a pressure cooker. All slides were then blocked in PBS with either 5% normal goat serum, 1% bovine serum albumin, and 0.01% triton-X or 5% normal rabbit serum and 0.01% triton-X at room temperature for 1 hr. Primary antibodies were then added to the same blocking serums and applied to the slides overnight at 4°C. Slides were then washed three times in PBS containing 0.1% Tween 20 for 10 min. After this, biotin conjugated secondary antibodies were added to the previous blocking solutions and applied to the slides for 1 hr at room temperature. Slides were washed for 10 min three times in PBS-Tween after which, Vectastain ABC reagent was applied to the slides for 30 min at room temperature. Slides were then washed for 10 min three times in PBS-Tween after which, Perkin Elmer Tyramide signal amplification, Cyanine-3 reagent was applied to the slides for 6 min at room temperature. Slides were then washed for 10 min three times in PBS-Tween and are then mounted using ProLong gold antifade mountant with Dapi. Slides were visualized using a Leica DMRX A2 upright light microscope and photographs were captured using LAS (Leica Application Suite) software package v4.2.0.

Primary antibodies used for IF were: rabbit polyclonal anti-CDX2 (Cell Signaling 3977, RRID:AB_2077043) at a 1:50 dilution, rabbit polyclonal anti-CEBPB (Santa Cruz Antibodies sc-150, RRID:AB_2260363) at a 1:2500 dilution, goat polyclonal anti-GATA3 (Santa Cruz Antibodies sc-22206, RRID:AB_2108588) at a 1:950 dilution, mouse monoclonal anti-GCM1 (abcam ab88748, RRID:AB_2041368) at a 1:200 dilution for placental tissue and 1:100 for mammary gland tissue, mouse monoclonal anti-Pan-Cytokeratins (AE1/AE3) (BioLegend, 914201, RRID:AB_2565152) at a 1:5000 dilution, and mouse monoclonal anti-Vimentin (Sigma v5255, RRID:AB_477625) at a 1:1000 dilution. Secondary antibodies used for IF were: biotinylated goat anti-mouse igG (Jackson Immunoresearch, 111-065-166, RRID:AB_2338569) at a 1:1000 dilution, biotinylated goat anti-rabbit igG (Jackson Immunoresearch, 111-065-144, RRID:AB_2337965) at a 1:5000 dilution, and biotinylated rabbit anti-goat igG (Vector, BA-5000, RRID:AB_2336126) at a 1:2500 dilution.

## Acknowledgements

We thank Jessica Chang and Qin Li for helpful discussions about bioinformatic analyses. We thank members of the wallaby research group and Professor Geoff Shaw in particular for help with collection and shipment of tammar tissues. We also thank the laboratory of Susan Fisher for the generous gift of Vimentin antibody. Genomic data has been deposited in the Gene Expression Omnibus, accession number: GSE90838. This work was supported by a National Science Foundation Graduate Research Fellowship (MWG; 2014185511), a Berry Foundation Postdoctoral Fellowship (GC), and grants from the Australian Research Council (MBR).

## Additional information

### Funding

| Funder | Grant reference number | Author |
|---|---|---|
| National Science Foundation | Graduate Research Fellowship 2014185511 | Michael W Guernsey |
| The Berry Foundation | Postdoctoral Research Fellowship | Guillaume Cornelis |
| Australian Research Council | Research Project Grant | Marilyn B Renfree |

The funders had no role in study design, data collection and interpretation, or the decision to submit the work for publication.

## Author contributions

MWG, Conceptualization, Formal analysis, Funding acquisition, Validation, Investigation, Visualization, Writing—original draft, Writing—review and editing; EBC, Conceptualization, Investigation, Writing—review and editing; GC, Data curation, Formal analysis, Writing—review and editing; MBR, Conceptualization, Resources, Project administration, Writing—review and editing; JCB, Conceptualization, Resources, Formal analysis, Supervision, Project administration, Writing—review and editing

## Author ORCIDs

Michael W Guernsey, http://orcid.org/0000-0002-7704-3039
Edward B Chuong, http://orcid.org/0000-0002-5392-937X
Guillaume Cornelis, http://orcid.org/0000-0002-4465-8328
Marilyn B Renfree, http://orcid.org/0000-0002-4589-0436
Julie C Baker, http://orcid.org/0000-0003-4057-9216

# Additional files

## Supplementary files

• Supplementary file 1. Genes, transcripts, and GO terms for differential expression analysis Excel spreadsheet containing (1) Up-regulated gene names for each category analyzed by differential expression (2) Up-regulated transcript IDs for each category analyzed by differential expression (3) GO terms for Up-regulated BOM genes (4) TOM genes (5) early genes and (6) late genes. GO terms discussed in the manuscript are highlighted in yellow.

• Supplementary file 2. Genes shared by the tammar, term mouse, and term human placentas Excel file containing all genes shared and uniquely expressed in the placenta of the marsupial, tammar wallaby, and eutherians, mouse and human.

• Supplementary file 3. Genes and GO terms for organ expression comparisons Excel file containing (1) All genes shared and uniquely expressed between tammar placenta, eutherian placenta, tammar mammary gland, and mouse mammary gland (analyzed categories highlighted) (2) Genes analyzed from placenta to liver and placenta to testis comparisons (3) GO terms for genes shared between tammar placenta, eutherian placenta, and tammar mammary gland (4) GO terms for genes shared between tammar placenta, eutherian placenta, and tammar liver (5) GO terms for genes shared between tammar placenta, eutherian placenta, and tammar testis (6) GO terms for genes shared only by eutherian placenta and tammar mammary gland (7) GO terms for genes shared only by eutherian placenta and tammar liver (8) GO terms for genes shared only by eutherian placenta and tammar testis. GO terms discussed in the manuscript are highlighted in yellow.

## Major datasets

The following dataset was generated:

| Author(s) | Year | Dataset title | Dataset URL | Database, license, and accessibility information |
|---|---|---|---|---|
| Guernsey MW, Baker JC | 2017 | Transcriptome sequencing in the tammar wallaby unveils conserved gene expression networks underlying reproduction in therian mammals | https://www.ncbi.nlm.nih.gov/geo/query/acc.cgi?acc=GSE90838 | Publicly available at the NCBI Gene Expression Omnibus (accession no: GSE90838) |

The following previously published datasets were used:

| Author(s) | Year | Dataset title | Dataset URL | Database, license, and accessibility information |
|---|---|---|---|---|
| Metsalu T, Viltrop | 2014 | Using RNA sequencing for | https://www.ncbi.nlm. | Publicly available at |

| Authors | Year | Title | URL | Availability |
|---|---|---|---|---|
| T, Tiirats A, Raja-shekar B, Reimann E, Kõks S, Rull K, Milani L, Acharya G, Basnet P, Vilo J, Mägi R, Metspalu A, Peters M, Haller-Kikkatalo K, Salu-mets A | | identifying gene imprinting and random monoallelic expression in human placenta (RNA-seq) | nih.gov/geo/query.acc.cgi?acc=GSE56524 | the NCBI Gene Expression Omnibus (accession no: GSE56524) |
| Fu NY, Lun A, Smyth GK, Visvader JE | 2015 | Transcriptome analysis of luminal and basal cell subpopulations in the lactating versus pregnant mammary gland | https://www.ncbi.nlm.nih.gov/geo/query.acc.cgi?acc=GSE60450 | Publicly available at the NCBI Gene Expression Omnibus (accession no: GSE60450) |
| Knox K, Baker JC | 2008 | Timecourse of developing mouse placenta, with placental and decidual tissues profiled separately | https://www.ncbi.nlm.nih.gov/geo/query.acc.cgi?acc=GSE11220 | Publicly available at the NCBI Gene Expression Omnibus (accession no: GSE11220) |
| Gingeras T | 2012 | File summary for ENCFF001IVS (fastq) | https://www.encodeproject.org/files/EN-CFF001IVS/ | Publicly available at ENCODE (accession no: ENCSR000BYQ) |
| Gingeras T | 2012 | Mus musculus C57BL/6 placenta adult (8 weeks) | https://www.encodeproject.org/experiments/ENCSR000BZP/ | Publicly available at ENCODE (accession no: ENCSR000BZP) |
| Cortez D, Marin R, Froidevaux L, Liechti A, Waters PD, Grützner F, Kaessmann H | 2014 | Origins and functional evolution of Y chromosome gene repertoires across the class Mammalia | https://www.ncbi.nlm.nih.gov/geo/query.acc.cgi?acc=GSE50747 | Publicly available at the NCBI Gene Expression Omnibus (accession no: GSE50747) |
| Snyder M | 2014 | Mus musculus C57BL/6 liver female adult (10 weeks) | https://www.encodeproject.org/experiments/ENCSR216KLZ/ | Publicly available at ENCODE (accession no: ENCSR216KLZ) |
| Gingeras T | 2012 | Mus musculus C57BL/6 liver adult (8 weeks) | https://www.encodeproject.org/experiments/ENCSR000BYS/ | Publicly available at ENCODE (accession no: ENCSR000BYS) |
| Snyder M | 2014 | Mus musculus C57BL/6 testis male adult (10 weeks) | https://www.encodeproject.org/experiments/ENCSR266ESZ/ | Publicly available at ENCODE (accession no: ENCSR266ESZ) |
| Gingeras T | 2012 | Mus musculus C57BL/6 testis male adult (8 weeks) | https://www.encodeproject.org/experiments/ENCSR000BYW/ | Publicly available at ENCODE (accession no: ENCSR000BYW) |

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
