## [Decision Letter]

Thank you for submitting your article "The tammar wallaby has a eutherian-like placenta that evolved as a molecular tradeoff between lactation and placentation" for consideration by *eLife*. Your article has been reviewed by three peer reviewers, one of whom, Antonis Rokas (Reviewer #1) is a member of our Board of Reviewing Editors and the evaluation has been overseen by Diethard Tautz as the Senior Editor.

The reviewers have discussed the reviews with one another and the Reviewing Editor has drafted this decision to help you prepare a revised submission.

Summary:

Guernsey and coworkers harness the power of a diverse set of comparative and functional genomic approaches and tools to examine the molecular program of the placenta and mammary gland of the tammar wallaby in the context of what is known about the molecular programs of the corresponding organs in eutherian mammals, like mice and humans. In particular, the authors present the fascinating hypothesis that gene expression has converged between eutherian placentas and marsupial mammary glands. To support this hypothesis they compare gene expression patterns in various extra embryonic tissues and mammary glands within wallaby and between wallaby and mouse. The analyses are divided into three parts. In parts one and two, the authors show that there are differences in gene expression between different parts of the wallaby placenta and that there is conservation of core gene expression between marsupial and eutherian placentas (particularly at the early stages of the eutherian placenta). In part three, the authors explore the similarity of gene expression, and thus function, between the marsupial mammary gland and eutherian placenta. They show that there is overlapping gene expression, and argue that the overlap implies convergent functional evolution for fetal development.

Essential revisions:

1) The authors explore the similarity of gene expression, and thus function, between the marsupial mammary gland and eutherian placenta. They show that there is overlapping gene expression, and argue that implies convergent functional evolution for fetal development. However, it is not possible to make this conclusion from the data because there is no null model specified. For example, how many genes would be expected to be expressed in the eutherian placenta, tammar placenta, and tammar mammary gland, but not in the mouse mammary gland based on a neutral model of gene expression evolution? Is the identification of 77 transcripts with that specific expression pattern more or less than one would expect based under the null model? The same argument applies to the tammar mammary gland, eutherian placenta, and mouse mammary gland, but not in the tammar placenta data. The authors need to compare other tissue sets to infer if what they observed for the placenta/mammary gland data is specific to that tissue pair or whether similar results would be found for example in placenta/brain, placenta/testis, and mammary gland/lung comparisons. One possible way to do this would be to compare patterns of gene expression sharing (as they have done for placenta, mammary gland, and heart) across additional tissue types by building a matrix (of tissue x gene expression) and showing that mammary gland/placenta have higher correlations of gene expression than either of those two tissues do with any other tissue. Alternatively, the authors could construct a network based similarity of gene expression across different tissues and show that the edges in the network are stronger between placenta and mammary gland than across other tissues. If the similarity between placenta and mammary gland is not different than for other pairs, the authors' observations are not specific to placenta/mammary gland and thus cannot be related to their hypothesis that gene expression has converged between eutherian placentas and marsupial mammary glands.

2) The authors infer that tammar wallabies have a "eutherian-like" placenta. But what does an "eutherian-like" placenta look like? And why isn't the inference that the mouse placental program isn't "marsupial-like"? And how does finding shared genes underlying both justify the claim that the wallaby has a eutherian-like placenta? We suggest that the authors should focus on the similarities and differences between the programs compared and avoid the labeling of the programs as fitting the eutherian or the marsupial patterns. In fact, that's exactly the problem the authors are trying to solve, which is that referring to eutherians as "placentals" and to marsupials as "non-placentals" is misleading. There are similarities and differences in the developmental programs of pregnancy in both lineages and it's important that the authors focus on those, rather than on simplified abstractions.

3) The authors infer that there is a molecular trade-off between lactation and placentation. For example, the authors explain that (subsection “Placenta and mammary gland tradeoffs in the evolution of therian mammals”) "…the genes shared between these tissues are enriched for processes involved in nutrition, immune function and embryonic development, all suggesting a direct tradeoff between placentation and lactation in mammalian evolution." The reviewers think the fact that genes are shared between lactation and placentation need not reflect trade-offs, and conversely, trade-offs between the two could be present in the absence of a shared gene set. For example, it could simply be the case that the program typically observed in a eutherian placenta has been simply partitioned into the placental and lactation marsupial programs. Trade-off implies compromise, which the authors have not demonstrated. Furthermore, the reviewers don't think the argument is sufficiently developed that a decisive test of the presence or absence of a trade-off can be made with these data. Importantly, the reviewers think that the authors don't even need to make a trade-off argument in the interpretation of their core results, and their generality, so the reviewers' suggestion is that the authors completely remove the argument.

4) Given comments #2 and #3, the authors should come up with a new title.

---

## [Author Response]

*Essential revisions:*

*1) The authors explore the similarity of gene expression, and thus function, between the marsupial mammary gland and eutherian placenta. They show that there is overlapping gene expression, and argue that implies convergent functional evolution for fetal development. However, it is not possible to make this conclusion from the data because there is no null model specified. For example, how many genes would be expected to be expressed in the eutherian placenta, tammar placenta, and tammar mammary gland, but not in the mouse mammary gland based on a neutral model of gene expression evolution? Is the identification of 77 transcripts with that specific expression pattern more or less than one would expect based under the null model? The same argument applies to the tammar mammary gland, eutherian placenta, and mouse mammary gland, but not in the tammar placenta data. The authors need to compare other tissue sets to infer if what they observed for the placenta/mammary gland data is specific to that tissue pair or whether similar results would be found for example in placenta/brain, placenta/testis, and mammary gland/lung comparisons.*

We very much appreciate reviewers’ insight and agree that control datasets would make the analysis more robust. Therefore, we used tammar liver and tammar testes transcriptomes as a “background’ set. We filtered the liver and testis data in the same manner described in the “comparative genomics” section of our methods revealing 7,506 genes expressed in tammar liver, 12,159 genes expressed in mouse liver, 8,094 genes expressed in tammar testis, and 12,249 genes expressed in mouse testis. We compared placenta transcriptomes to liver transcriptomes and then to testis transcriptomes to examine the genes shared in analogous categories to our mammary gland analysis.

First Analysis: Genes expressed in tammar placenta, eutherian placenta, tammar mammary gland, but not in mouse mammary gland (77 genes, Figure 5), including GCM1 and genes associated with nutrient transport (p-adj= 0.01627123), which given the small number of genes in this group is surprising.

Liver Analysis: We found 76 genes expressed in tammar placenta, eutherian placenta, tammar liver, but not in mouse liver. While this number is similar to that in the first analysis, it was never our intention to imply that the specific number of genes in each category was significant, but rather that the *specific identity of genes and their associated functions was important.* Keeping this in mind, we find no significant GO terms associated with biological processes nor do we find expression of GCM1.

Testis Analysis: We found 589 genes expressed in tammar placenta, eutherian placenta, tammar testis, but not in mouse testis. We find no significant GO terms associated with biological processes, which is surprising due to the larger number of genes in this category, nor do we find expression of GCM1.

We have added the control analysis to the paper.

Second Analysis: Genes expressed in eutherian placenta, tammar mammary gland, mouse mammary gland, but not tammar placenta (1,915 genes, Figure 5), including GATA3 and genes associated with embryonic development.

Liver Analysis: We found genes expressed in the eutherian placenta, tammar liver, mouse liver, but not in tammar placenta. Here we find 2,749 genes expressed. We do not find GATA3 expressed here, however we do find significant GO terms associated with embryonic development (p-adj=2.56 X 10^-7^).

Testis Analysis: We found genes expressed in the eutherian placenta, tammar testis, mouse testis, but not in tammar placenta. Here we find 2,314 genes expressed. We find both GATA3 expressed here and the presence of significant GO terms associated with embryonic development (p-adj=1.55 X 10^-6^).

*As both background sets identifying similar significant function to the original analysis, we cannot say that genes in this category represent a functional relationship between placentation and lactation and, as such, we have opted to remove discussion of this category from the manuscript. We have still retained the GATA3 immunofluorescence as it is a gene with important placental function that exhibits expression in both mouse and tammar mammary glands.*

Third Analysis: Genes expressed in eutherian placenta and tammar mammary gland, but not in tammar placenta or mouse mammary gland (108 genes, Figure 5), including IGFBP1 and genes associated with embryonic morphogenesis and immune function.

Liver Analysis: We found 168 genes expressed in the eutherian placenta, tammar liver, but not in the tammar placenta or mouse liver. We find only GO terms associated with neural processes (p-adj= 0.011296591) and Insulin-like growth factors are not included in this gene group.

Testis Analysis: We found 788 genes expressed in the eutherian placenta, tammar testis, but not in the tammar placenta or mouse testis. The biological functions associated with this group of genes include nucleic acid metabolism (p-adj= 0.000599), cell cycle (p-adj= 0.0494) and nuclear transport (p-adj= 0.0428). We do not find any Insulin-like growth factors expressed in this category.

We have added the control analysis to the paper.

Genes and GO terms for all analysis have been added as lists in “[Supplementary-material SD3-data]”.

*One possible way to do this would be to compare patterns of gene expression sharing (as they have done for placenta, mammary gland, and heart) across additional tissue types by building a matrix (of tissue x gene expression) and showing that mammary gland/placenta have higher correlations of gene expression than either of those two tissues do with any other tissue. Alternatively, the authors could construct a network based similarity of gene expression across different tissues and show that the edges in the network are stronger between placenta and mammary gland than across other tissues. If the similarity between placenta and mammary gland is not different than for other pairs, the authors' observations are not specific to placenta/mammary gland and thus cannot be related to their hypothesis that gene expression has converged between eutherian placentas and marsupial mammary glands.*

We compared patterns of gene expression sharing between placenta, mammary gland, testes and liver in the two species, using the same strategy as in Figure 4. First, we find that the tammar and mouse placenta share the highest correlation of any cross-species organ pair (placenta 0.31, mammary gland 0.03, liver 0.06, testis 0.04; Figure 5—figure supplement 1), confirming the high degree of molecular similarity between the eutherian and tammar placenta that we found using immunofluorescence (see Figure 3). Second, the tammar mammary gland shows the highest correlation with the mouse mammary gland when compared to any other mouse tissue (placenta -0.17, mammary gland 0.03, liver -0.12, testis -0.06), highlighting some functional conservation in this organ. Finally, the placentas of both tammar and mouse show the lowest within species correlation with mammary gland, which suggests widely divergent roles for each organ in supporting reproductive physiology. While the reviewers are concerned that a lack of similarity might reflect total divergent roles for these organs in supporting reproductive physiology, we would argue that while clearly the two organs have exceptionally compartmentalized function, there is an overlap (77 genes that have nutrient and placental functions) that emerges in more targeted, less global, examination.

Overall, we thank the reviewers for the suggestion to complete this analysis throughout comment #1 because it bolsters our previous claim that the marsupial and eutherian placentas are similar and suggests, that the placenta and mammary gland have some shared functions that have been differently partitioned in the tammar. We have changed the language in the manuscript to emphasize the shared genes and functions of the placenta and mammary gland while making sure to avoid any language that asserts that there is something significant about the specific number of genes (i.e. 77) in each category.

*2) The authors infer that tammar wallabies have a "eutherian-like" placenta. But what does an "eutherian-like" placenta look like? And why isn't the inference that the mouse placental program isn't "marsupial-like"? And how does finding shared genes underlying both justify the claim that the wallaby has a eutherian-like placenta? We suggest that the authors should focus on the similarities and differences between the programs compared and avoid the labeling of the programs as fitting the eutherian or the marsupial patterns. In fact, that's exactly the problem the authors are trying to solve, which is that referring to eutherians as "placentals" and to marsupials as "non-placentals" is misleading. There are similarities and differences in the developmental programs of pregnancy in both lineages and it's important that the authors focus on those, rather than on simplified abstractions.*

We agree that the data do not allow us to infer whether the tammar has a “eutherian-like” placenta or the mouse has a “marsupial-like” placenta. We have honed in on the convergent functions of these two placentas rather than making distinctions about the directionality of evolutionary change.

*3) The authors infer that there is a molecular trade-off between lactation and placentation. For example, the authors explain that (subsection “Placenta and mammary gland tradeoffs in the evolution of therian mammals”) "…the genes shared between these tissues are enriched for processes involved in nutrition, immune function and embryonic development, all suggesting a direct tradeoff between placentation and lactation in mammalian evolution." The reviewers think the fact that genes are shared between lactation and placentation need not reflect trade-offs, and conversely, trade-offs between the two could be present in the absence of a shared gene set. For example, it could simply be the case that the program typically observed in a eutherian placenta has been simply partitioned into the placental and lactation marsupial programs. Trade-off implies compromise, which the authors have not demonstrated. Furthermore, the reviewers don't think the argument is sufficiently developed that a decisive test of the presence or absence of a trade-off can be made with these data. Importantly, the reviewers think that the authors don't even need to make a trade-off argument in the interpretation of their core results, and their generality, so the reviewers' suggestion is that the authors completely remove the argument.*

We agree that the argument that a “tradeoff” has occurred between placentation and lactation distracts from the major points of the manuscript. We have revised this argument to state that these two processes have evolved some convergent function despite the great differences in reproductive strategy.

*4) Given comments #2 and #3, the authors should come up with a new title.*

We agree that the previous title does not reflect the major points of the manuscript and have changed the title to “Molecular conservation of marsupial and eutherian placentation and lactation.” to better represent our arguments.